# Sub-ps Pulsed Laser Deposition of Boron Films for Neutron Detector Applications

**DOI:** 10.3390/ma16041512

**Published:** 2023-02-11

**Authors:** Maria Luisa De Giorgi, Muhammad Rizwan Aziz, Alexandra Manousaki, Alessio Perrone, Argyro Klini

**Affiliations:** 1Dipartimento di Matematica e Fisica “E. De Giorgi”, Università del Salento, 73100 Lecce, Italy; 2INFN-Istituto Nazionale di Fisica Nucleare, 73100 Lecce, Italy; 3Institute of Electronic Structure and Laser (IESL), Foundation for Research and Technology-Hellas (FORTH), 71110 Heraklion, Greece

**Keywords:** sub-ps pulsed laser deposition, boron thin films, neutron converters, thermal neutrons, average ablation rate, average ablation efficiency

## Abstract

In view of the demand for high-quality thermal neutron detectors, boron films have recently attracted widespread research interest because of their special properties. In this work, we report on the deposition of boron films on silicon substrates by sub-picosecond pulsed laser deposition (PLD) at room temperature. Particular emphasis was placed on the investigation of the effect of the laser energy density (fluence) on the ablation process of the target material, as well as on the morphological properties of the resulting films. In addition, based on the study of the ablation and deposition rates as a function of the fluence, the ablation/deposition mechanisms are discussed. We show that well-adherent and stable boron films, with good quality surfaces revealing a good surface flatness and absence of cracks, can be obtained by means of the PLD technique, which proves to be a reliable and reproducible method for the fabrication of thick boron coatings that are suitable for neutron detection technology.

## 1. Introduction

Boron (B) and covalently bonded boron compounds (e.g., B_x_O, B_4_C, BN) were proposed for application in various technological fields, thanks to their chemical, tribological, physical and nuclear properties [1,2,3,4,5,6,7,8]. In particular, boron thin films are widely studied in the field of neutron detection, which represents one of its main applications, due to the high reaction cross-section of boron for thermal neutron capture [9,10,11].

Neutron detection and reliable determination and efficiency are of great importance, due to their applications in industry, medicine and science.

Currently, neutron detector devices are based on elements such as ^6^Li, ^10^B and ^3^He, which have high reaction cross-sections for thermal neutrons (0.025 eV) of about 940, 3840 and 5330 barns, respectively. These high cross-sectional values make them strong candidates for neutron detection systems, even in the case of low thermal neutron flux.

So far, the ^3^He isotopic species (natural abundance: 1.37 ppm) is widely used for the detection of neutrons in the above-mentioned applications because of its favorable properties, such as inertness, stability and weak sensitivity to other types of radiation. However, the increased use of ^3^He in the last few years has led to a remarkable shortage; hence, research interest was focused on the development of alternative technologies that are capable of substituting helium-based systems.

As an alternative converter material in neutron detectors, ^10^B appears to be an attractive candidate due to its relatively high natural isotopic abundance (19.9% at.) and its high reaction cross-section for thermal neutrons’ beams [10,12,13,14,15]. Neutron detection through boron is based on the nuclear reaction ^10^B(*n*,α)^7^Li, where an incident neutron beam reacts with boron atoms, producing electrically charged particles (α and ^7^Li particles) that, in turn, produce a detectable electrical signal. However, its high evaporation temperature and the high chemical reactivity of boron with oxygen and oxygen-containing molecules (H_2_O and CO_2_) have prevented the deposition of high-quality boron thin films [1] in terms of thickness uniformity, compactness and adhesion and in turn has prevented their application in neutron detection.

In the literature, different approaches have been reported for the fabrication of boron thin films, even though the available chemical and physical methods have not been fully explored. Focusing on physical methods [16], they possess significant benefits compared to their chemical counterparts. Not only do they require low temperatures, thus facilitating minimum power consumption, but they also avoid the use of expensive and chemically hazardous toxic and/or explosive gases [17,18].

It has been shown that vacuum arc deposition [3] allows the growth of compact boron coatings, due to the high kinetic energy of the depositing species. However, the converter surface appears to be non-uniform, with some areas covered with macro-particles and other areas left uncoated, as a result of the arc voltage fluctuations and the random spatial distribution of the generated ions [6]. Additionally, the magnetron sputtering technique, while being suitable for large-area coating [19], has been proven to be inappropriate for the deposition of boron films [20] because pure boron atoms are scarcely expelled by sputtering due to the low atomic number of this element [21].

Moreover, high temperatures are required to improve film adhesion [2,14,18,22], which, along with low-pressure conditions [23], is a parameter that affects the structure and density of the deposited films.

Among the physical vapor deposition techniques, pulsed laser deposition is a well-established, versatile deposition technique with considerable advantages with respect to the above-mentioned techniques. It is a clean, straightforward and flexible deposition scheme [24,25], compatible with different type of materials, which permits the deposition of uniform films with good adhesion and weak residual stress [10,26,27,28].

Most studies reported in the literature employ nanosecond-pulsed [1,29] laser sources in the ultraviolet region, although studies employing femtosecond pulses have also been carried out. The motivation of these studies is to exploit potential advantages related to ultrafast laser ablation, such as non-thermal melting, highly energetic ablated species and the absence of interaction between the incoming laser beam with the expanding plume, thereby improving the film quality and its properties.

Indeed, the PLD technique in the nanosecond range presents some drawbacks; the ns-process is characterized by a low deposition rate (around 30 nm/min [1]) and the produced films are composed of micro- and nanoaggregates and particulates, giving rise to high roughness on surfaces, which is particularly unfavorable for neutron converters.

Nevertheless, deposition of boron films with sub-ps laser irradiation is reported in the literature only in the work of Costa et al. [2]. The films, deposited with a 25 fs laser in the near infrared (@ 775 nm), are scarcely close-packed with a flaky morphology, whereas, for neutron conversion applications, high density is necessary because it implies a higher detection efficiency. Moreover, they present an amorphous structure with a thickness uniformity just over a few mm^2^.

In the work presented in this paper, we report on the effectiveness of ultrashort-pulsed laser ablation and deposition of boron at room temperature, using UV sub-picosecond pulses. In particular, we report a systematic investigation of the influence of laser fluence, ranging from 0.4 to 3.3 J/cm^2^, on the deposition process and on the quality of the resulting films. Additionally, we have explored the physics that characterizes both the ablation and deposition processes.

The fabricated boron films are characterized by close-packed growth, with uniform surface morphology and excellent adherence to the Si substrate.

Finally, the average ablation and deposition rates imply that sub-ps PLD is an efficient technique for obtaining boron films with a thickness that is appropriate for applications in the conversion of the thermal neutrons technology within a reasonable time.

## 2. Materials and Methods

PLD experiments were carried out by using boron targets with natural isotopic abundances (19.9% of ^10^B and 80.1% of ^11^B). Since the aim of this study is the characterization of the surface morphology and the assessment of the thickness uniformity of the obtained films and their adherence to the substrate, and since the chemical properties of ^10^B-enriched targets are the same as those of the targets with natural isotopic abundances, the use of the latter is justified. The boron thin films were deposited on Si substrates (Goodfellow GmbH Supplier) at room temperature in a conventional stainless steel high vacuum PLD apparatus (background pressure of 2 × 10^−5^ Pa), which was described in more detail in a previous paper [30]. For the deposition of boron thin films, a hybrid dye-excimer laser (Lambda Physik EMG 150), with a pulse duration of 0.5 ps and a wavelength of 248 nm, was focused on the boron target surface by a 30 cm focal length quartz lens. The boron target was rotated at a frequency of 1 Hz to avoid deep crater formation [31], and the incident angle of the laser beam on the target surface was 45°. The silicon substrate was placed at a target–substrate distance of d = 3 cm. Prior to each deposition, the target surface was cleaned with 1000 laser pulses in order to remove contaminants (mainly oxides and adventitious carbon). The deposition processes were carried out with a total number of 40.000 laser pulses, with the laser operating at 10 Hz. The laser fluence, calculated as the ratio of the laser energy impinging onto the target surface relative to the irradiated area, was varied in the range between 0.4 and 3.3 J/cm^2^ by changing the laser pulse energy (using a variable dielectric attenuator) and keeping the laser spot area constant, with an accuracy of 20%. Laser power density was in the range 0.8–6.7 TW/cm^2^.

In order to determine the film thickness profile, the Si substrates were covered with copper mesh consisting of wire that was 200 μm in diameter and had an aperture of 1.5 mm. To avoid material deposition under the metal wire, the mesh was mechanically fixed onto the substrate by using a mask and screws, ensuring close contact between the substrate and mesh. Deposition through this grid resulted in a 2D array of spots. The film thickness was measured using a profilometer (Tencor Alphastep), which was used to deduce the average deposition rate. The average ablation rate (AR) was measured by weighing the target prior and after the ablation process using a digital scale, with an instrumental uncertainty of 0.01 mg.

The morphology of the deposited films was investigated through Scanning Electron Microscopy (SEM, JEOL JSM-6390LM). The chemical composition of the films was deduced by EDX analysis (OXFORD INCAPenta FETx3). The accuracy of the measurements was about 15%.

The adhesion of the boron film deposited on the Si substrate was evaluated by the tape test (known as the “Scotch-tape method”) [32].

The surface reflectance of the boron target was experimentally determined by means of a UV-VIS Lambda 900 Perkin-Elmer spectrophotometer, equipped with a Labsphere integrating sphere.

## 3. Results and Discussion

It is well known that the laser-induced ablation process is strongly dependent on the laser parameters (pulse duration, wavelength, laser fluence), on the optical properties (reflectivity, absorption), and on the thermophysical properties (density, thermal conductivity, specific heat) of the target material. To assess how laser fluence impacts on the quality of the fabricated film, both the evaporation/ablation and the expansion of the ablated material were analyzed in a series of experiments at fluence values ranging from 0.4 to 3.3 J/cm^2^.

### 3.1. Ablation Process: Average Ablation Rate and Ablation Efficiency

In order to study the effectiveness of the sub-ps deposition process of B, we first focused on the ablation phenomenon induced by the laser-target interaction. Quantitatively, it is described by the ablation threshold fluence and average ablation rate.

The experimental average ablation rate at different fluences was deduced through the target mass loss by weighing it before, m_i_, and after, m_f_, the ablation process.

The mass loss of the target, Δm = m_i_ − m_f_, following the laser ablation with different energy fluences was first determined. From these values, the average ablation rate (defined as Δm per pulse) and the average ablation efficiency (defined as the average ablation rate per unity of energy) were calculated. The corresponding values are presented in Table 1.

Figure 1a illustrates the average ablation rate values of the boron target as a function of the laser pulse fluence and the power density. As evidenced, the increase in fluence to 3.3 J/cm^2^ led to an approximately logarithmic increase in the average ablation rate of up to 33.2 ng/pulse. The fitting curve is described by the following equation [33]:(1)$$AR=A\ln\left(\frac{F}{F_{th0}}\right)$$

In the fitting procedure, both *A* and *F_th0_* are free parameters. *A* is related to the ablation efficiency and proportional to the optical absorption coefficient, while *F*_*th*0_ represents the fluence corresponding to the start of the ablation process (below this value, *AR* = 0). *A* is equal to 6.53 ± 0.08 ng/pulse while *F*_*th*0_ is equal to 0.022 ± 0.006 J/cm^2^.

The efficiency of the ablation process directly relates to the minimum fluence at which appreciable material removal is observed (i.e., the laser fluence threshold, *F_th_)* and the amount of material evaporated per pulse at *F_th_, m(F_th_)*.

According to the model of Singh and Narayan [34], *F_th_* can be theoretically deduced by the following formula [30]:(2)$$F_{th}=\frac{\rho (L_{T}+L_{O})}{3}\left(\frac{c_{S}\Delta T+\Delta H_{f}+\Delta H_{e}}{1-R}\right)$$

In this formula, *ρ* is the density of the target, *c_s_* is its specific heat, Δ*T* is the difference between the boiling temperature (4200 K) and room temperature, Δ*H_f_* is the molar latent heat of fusion, Δ*H_e_* is the molar latent heat of vaporization and *R* is the surface reflectance of the target at the laser wavelength of 248 nm. *L_o_* represents the optical absorption length, L0=(2ωμ0σ)1/2, where *ω* is the radiation pulsation, *μ_0_* is the magnetic permeability in vacuum and *σ* is the electrical conductivity.

The thermal diffusion length, *L_T_*, is given by the equation $L_{T}={\left(\alpha \tau \right)}^{1/2},\;$where α=kρcS is the thermal diffusivity, *k* is the thermal conductivity and *τ* is the laser pulse duration. The mass, optical and thermal properties of the boron target are reported in Table 2.

The material evaporated per pulse at the ablation threshold, *m(F_th_)*, depends on the optical and thermal properties of the target, as well as on the characteristics of the impinging laser radiation. Its value can be calculated through the following relation [33]:(3)$$m\left(F_{th}\right)=\rho \frac{S}{3}\left(L_{T}+L_{O}\right)$$
which results in about 10.6 ng/pulse.

From the fitted logarithmic curve (Figure 1a, dashed line), the experimental value of the evaporated mass, $m\left(F_{th}\right)$, at the threshold laser fluence value (0.08 J/cm^2^) is about 8.6 ng/pulse.

To gain more insight into the ablation and deposition processes, the behavior of the laser ablation efficiency as a function of the laser fluence and power density was studied. The experimental data are shown in Figure 1b. In accordance with previous experimental observations on metallic materials [35], at a low laser fluence regime (below 1 J/cm^2^), the curve exhibits a typical negative exponential trend (dashed red fitting curve in the figure). The unusual increase in the ablation efficiency observed at 3.3 J/cm^2^ could be explained by the transition of the ablation process from the spallation regime to the phase explosion regime [36,37]. In ablation processes of metal targets, the phase explosion regime occurs at relatively high fluence. As the fluence exceeds a critical value, a fast transition from an overheated liquid to a vapor phase begins. The transition between the two regimes as a potential ablation mechanism in metals by irradiation of a sub-ps laser is characterized by the presence of flakes and particulates on the film surface, which was also indicated in recent research studies by Li and Guan [38]. The same evidence was observed in our current work at 3.3 J/cm^2^.

This topic will be taken up later in Section 3.2.2.

### 3.2. Deposition Process and Film Characterization

#### 3.2.1. Deposition Rate and Film Thickness

We investigated the role of the fluence on the film thickness and analyzed its uniformity on the substrate.

The thickness (Th) of the deposited film was not perfectly uniform across the substrate, and in particular, it decreased slightly towards the region farthest from the film center. Depending on the pulse fluence, the thickness across the film follows the cosine law, $cos^{n}\theta$, with *n* ≥ 1 (where *n* is related to the film flatness and *n* = 1 corresponds to a conventional thermal evaporation process). In Figure 2, the thickness of the deposited boron films is plotted for the samples deposited at fluences of 1.1 and 3.3 J/cm^2^ as a function of the θ angle from the ablation plume axis, and as a function of the corresponding distance, l, from the film center.

In these experiments, the non-uniformity of the film thickness was mainly attributed to the plume expansion geometry and the angular distribution of plasma particles, and to a lesser extent to the plume deflection effect [39,40].

The thickness experimental data as a function of θ, Th(θ), were fitted with the cosine law
(4)$$Th\;\left(\theta \right)=Th_{0}\times cos^{n}\theta \;$$
where $Th_{0}$ represents the thickness at the center of the deposited boron film.

As evidenced from the fitting curves of Figure 2a for the sample deposited at 1.1 J/cm^2^ (black dots and dashed fit curve), the value of $Th_{0}\;$is equal to 321 ± 11 nm and *n* = 5.0 ± 0.3, while for the sample deposited at 3.3 J/cm^2^ (red dots and dashed fit curve), the values are $Th_{0}$ = 240 ± 9 nm and *n* = 9.4 ± 0.9. These values imply that with increasing fluence, the film thickness decreases, but with a more peaked dome shape.

The thickness values of the deposited films, measured at 7 mm from the center of films (uncertainty around 10%), are reported as a function of the fluence in Table 1 and in Figure 2b. At low energy densities (ranging from 0.4 to 1.1 J/cm^2^), the thickness of the film shows a steep increase from 32 nm to 265 nm, as shown in Figure 2b. On the contrary, at 3.3 J/cm^2^, the thickness drops slightly. This unexpected decrease, in contrast with the trend that the average ablation rate increases with the laser fluence, could be explained with the onset of the plume deflection effect at a high laser fluence. Indeed, the presence of such an effect at a high laser fluence produces the growth of a non-uniform material deposition, as the ablated material is no longer directed to the center of the substrate [40].

This effect is considered one of the main disadvantages of the PLD technique.

#### 3.2.2. Film Surface Morphology and Composition

The surface morphology of the boron films deposited in the investigated laser fluence range was probed by SEM analyses (Figure 3).

The films present good surface flatness without cracks.

At laser fluences ranging from 0.4 to 0.8 J/cm^2^, the samples were characterized by the presence of sub-micrometer droplets, with diameters in the range between 100 and 300 nm, coming from the target during the ablation process. The circular shape suggests their formation into a liquid phase on the target surface. At a higher fluence (1.1 J/cm^2^), droplets, flakes and irregular fragments begin to appear, which are typical of films deposited with sub-ps lasers [2]. Finally, at 3.3 J/cm^2^ (Figure 3,A5), no droplets are detected and the surface is fully covered by flakes and fragments. The change in surface morphology is likely due to the variation of ablation processes with laser fluence and, in particular, confirms the transition from the spallation to the phase explosion regime.

The surface droplet density, by counting the number of the droplets over an area of about 120 μm^2^, was estimated. The results, reported in Table 1, are plotted in Figure 3 as a function of the laser fluence, ranging from 0.4 to 1.1 J/cm^2^. The graph shows that the surface droplet density grows with the fluence.

During the SEM observations, elemental analyses through EDX technique have also been performed. From the EDX spectra, we inferred that the deposited films were composed essentially of boron atoms. However, in the spectra, we also observed oxygen and carbon peaks, indicating that the surface was polluted by oxygen and carbon (with an atomic concentration below 10%), presumably coming from environmental and vacuum chamber residual gas.

#### 3.2.3. Film Adherence to Substrate and Stability

In our experimental conditions (10^12^–10^13^ W/cm^2^), the ions reached the substrate surface with velocities in the range of 10^6^–10^7^ cm/s [41]. The ions penetrate the substrate surface with a high kinetic energy, substantially realizing an ion implantation process and thereby firmly fixing the ablated material to the substrate [42]. High sticking quality, even on the flat Si wafer surface, was observed. The good adhesion of the films on the Si substrates was confirmed with the Scotch-tape test.

Furthermore, the good adhesion of our deposited films on the substrates was continuously monitored for some months after the deposition. In this period, the films were kept in open air, and they did not peel off from the substrate. Therefore, we can conclude that the adhesion of boron films prepared by sub-ps laser lasts for a long time, and it is not affected by exposure to the atmosphere.

As a result, the PLD technique in the sub-ps regime in peculiar conditions is particularly suitable for preparing boron films for use in thermal neutron detectors. Indeed, it must be emphasized again that this tribological property of the films is strongly demanding for any kind of application.

## 4. Conclusions

In this work, boron thin films were deposited on flat Si substrates in a high vacuum and at room temperature using the PLD technique. It was proven and confirmed that the high kinetic energy of the ablated material produced in our experimental conditions (10^12^–10^13^ W/cm^2^) leads to the production of boron films of good quality, with high adhesion to the substrate. Hence, the growth of boron thin films by PLD seems to be a very promising approach to overcome adhesion problems.

In the fluence range 0.4–1.1 J/cm^2^, the boron films were covered with sub-micrometer droplets with a circular geometrical shape and flattened morphology due to the fact that the ablated material is ejected in liquid phase from the target surface. Alternatively, the sample deposited at 3.3 J/cm^2^ showed a surface covered with flakes and fragments, likely due to the transition of the ablation process from the spallation regime to the phase explosion regime.

It has also been verified that the samples of boron thin films on Si substrates can stay in open air for several months before their potential use as thermal neutron detectors. The present achievements suggest to us that high-quality boron thin films, in terms of surface morphology, ablation and deposition rates, can be obtained by carrying out the ablation process at relatively low laser fluences (from 0.4 to 1.1 J/cm^2^), even at room temperature.

In the future, starting with the results obtained in this work, we are planning to deposit films using ^10^B-enriched targets to be tested as thermal neutron detectors. Indeed, even if the relatively high isotopic abundance of ^10^B in targets of boron with natural isotopic concentrations could warrant the use of the deposited films for neutron sensitive converters, ^10^B-enriched targets will certainly improve the detection efficiency.

## Figures and Tables

**Figure 1 materials-16-01512-f001:**
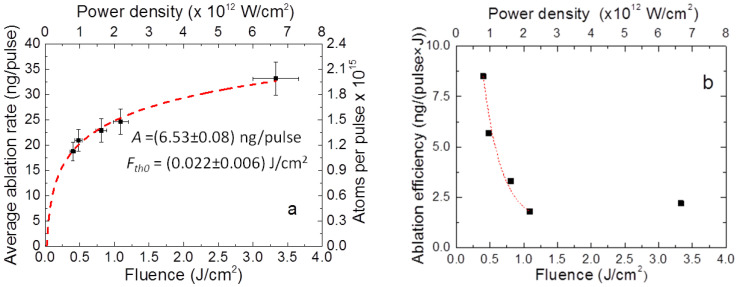
Fluence dependence of the (**a**) average ablation rate and number of boron atoms ablated per single pulse (black scatter points) with a fitting logarithmic curve (dashed red line) and (**b**) ablation efficiency (black scatter points) with a fitting exponential curve (dashed red line).

**Figure 2 materials-16-01512-f002:**
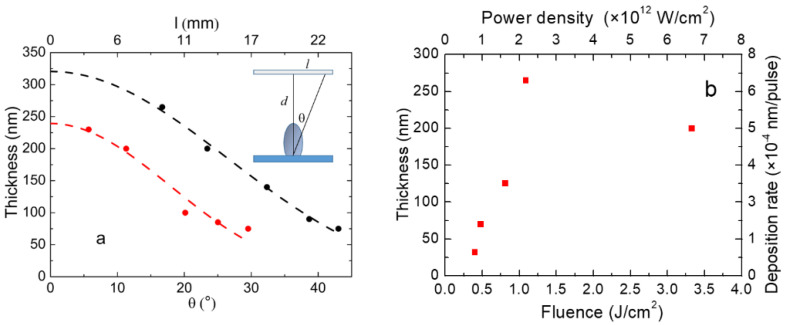
Thickness (*Th*) of the deposited boron films grown at room temperature as a function of the (**a**) *θ* angle from the ablation plume axis for the samples deposited at 1.1 (black dots) and 3.3 J/cm^2^ (red dots) and (**b**) the fluence at *l* = 7 mm (red squares).

**Figure 3 materials-16-01512-f003:**
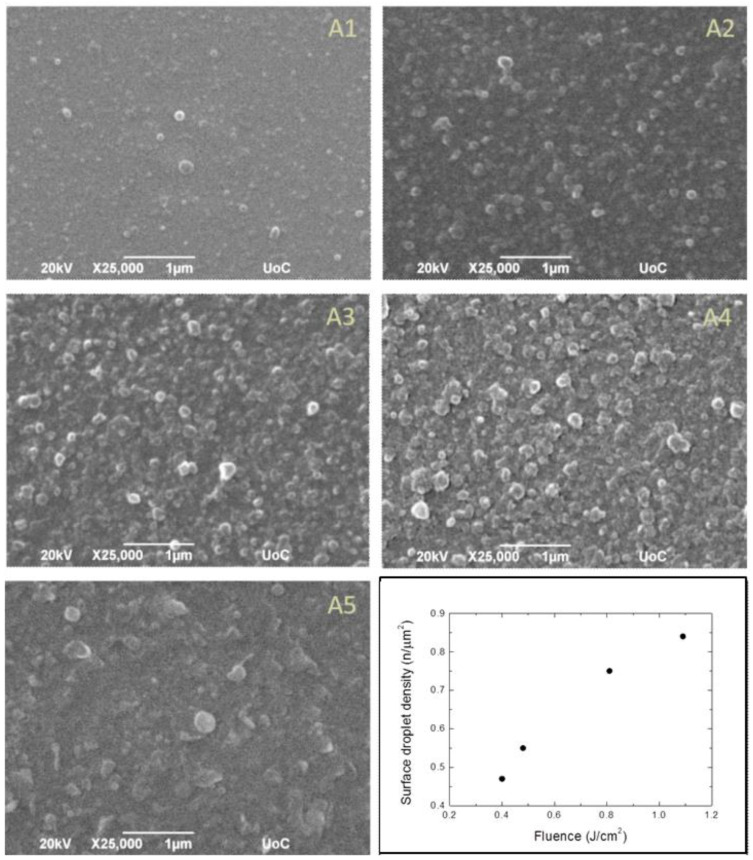
SEM micrographs of the deposited boron film substrates (**A1**–**A5**) and a plot of surface droplet density as a function of the laser fluence obtained from samples in **A1–A4**.

**Table 1 materials-16-01512-t001:** The table summarizes the parameters used during the pulsed laser deposition (beam energy and laser fluence), the quantities characterizing the laser ablation process (average ablation rate, number of atoms per pulse and average ablation efficiency) and the deposition process (film thickness at 7 mm from the film center, deposition rate and film surface droplet density).

Sample	A1	A2	A3	A4	A5
Energy (mJ)	2.2	3.7	6.9	13.6	15.3
Laser fluence (J/cm^2^)	0.4	0.5	0.8	1.1	3.3
Average ablation rate (ng/pulse)	18.8	21.0	22.9	24.6	33.2
Atoms/pulse (n/pulse) × 10^15^	1.05	1.17	1.28	1.37	1.85
Average ablation efficiency (ng/(pulse × J))	8.5	5.7	3.3	1.8	2.2
Thickness (nm)	32	70	125	265	200
Deposition rate (nm/pulse) × 10^−4^	8	18	31	66	50
Droplet density (n/μm^2^)	0.47	0.55	0.75	0.84	flakes

**Table 2 materials-16-01512-t002:** The mass, optical and thermal properties of the boron target. All the values are given for standard ambient temperature and pressure.

ρ (kg/m^3^)	2.37 × 10^3^
c_s_ (J/(kg × K))	1.02 × 10^3^
ΔT (K)	3907
ΔH_f_ (kJ/mole)	50.2
ΔH_e_ (kJ/mole)	500.8
k (W/(m × K))	27.4
α (m^2^/s)	11.3 × 10^−6^
σ (m^−1^Ω^−1^)	1.4 × 10^−4^
R@248 nm	0.09

## Data Availability

The data presented in this study are available on request from the corresponding author.

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
