# Peer review of "Sub-ps Pulsed Laser Deposition of Boron Films for Neutron Detector Applications"

_materials, 2023, doi:10.3390/ma16041512_

Round 1

Reviewer 1 Report

Authors have presented an interesting work in the area of the sub-ps laser deposition. The topic is relevant and timely, and the results are mostly clearly presented. Before publishing of this manuscript, I would suggest few corrections/improvements:

1. Some references in the introduction, where relevant previous and present research should be mentioned, are rather old. If possible, timely references regarding the new research of related topics should be added.

2. Lines 93 and 97: "2" in "mm2" and "J/cm2" should be written as superscript.

3. The sentence in lines 201-202 should be modified.

4. Lines 313-314: If the adhesion has been continuously monitored, the results in the form of, for example, chosen microscopic images could be presented.

Author Response

Dear Reviewer,

Please find the revisions to manuscript (materials-2164615 ) “Sub-ps pulsed laser deposition of boron films for neutron detector applications”, by Maria Luisa De Giorgi, et al.  

We thank you for your positive evaluation of the paper and in order to address the requested revisions we modified the paper in several points, which are evidenced in the manuscript in red colour.

Reviewer 1

Authors have presented an interesting work in the area of the sub-ps laser deposition. The topic is relevant and timely, and the results are mostly clearly presented. Before publishing of this manuscript, I would suggest few corrections/improvements:

Comment 1: Some references in the introduction, where relevant previous and present research should be mentioned, are rather old. If possible, timely references regarding the new research of related topics should be added.

Authors’ reply: We accepted the reviewer’s suggestion and updated the cited works. The new references are n. 3, 5, 9, 13-15, 18, 20, 21, 24, 26, 27 and 29.

Comment 2: Lines 93 and 97: "2" in "mm2" and "J/cm2" should be written as superscript.

Authors’ reply: The suggested modifications have been accomplished.

Comment 3: The sentence in lines 201-202 should be modified.

Authors’ reply: In order to address this comment, the sentence in consideration has been modified in “The mass, optical and thermal properties of the boron target are reported in Table 2.” Row 197

Comment 4: Lines 313-314: If the adhesion has been continuously monitored, the results in the form of, for example, chosen microscopic images could be presented.

Authors’ reply: We agree with the Reviewer 1 that it would have been nice to show the images of the aged films. We monitored the adhesion of the films on the Si substrates with the scotch-tape test and we took the image reported in the pdf file where the adhesion is well demonstrated. Nevertheless, we would rather not include this image in the paper for the sake of brevity and because it does not add any other information.

Best Regards

Maria Luisa De Giorgi

Reviewer 2 Report

The manuscript submitted by Giorgi et al. ("Sub-ps pulsed laser deposition of boron films for neutron detector applications") focuses on the deposition of boron films at room temperature on silicon substrate using the ps-PLD technique. The authors investigate the effect of laser fluence on the laser ablation process of the boron target and subsequently on the obtained film properties. The research is of interest since it addresses a current issue, namely the search for alternative methods that can be used in the manufacture of boron-based neutron detectors. However, the manuscript still needs a lot of work in terms of structuring, the presentation of additional experimental details, and lastly, the requirement for additional explanations of some of the obtained results. The remarks I would like to make are given below.

1. The manuscript contains some small inaccuracies that can be easily corrected such as: the "synthesis" term from the abstract should be replaced with deposition, fabrication, production, etc.; there is no consistency regarding the introduction and use of abbreviations (for example, the terms PLD, B film, fluence F from row 170 etc.); one single decimal should be used for the laser fluence estimates (“…laser fluence, ranging from 0.40 to 3.33 J/cm2”) and also specified in the experimental details section how the laser fluence was determined; the provider of the silicon substrate must be specified in experimental section; it is not necessary to mention all over the manuscript the fluence range, it can be simply stated "in the investigated fluency range" instead of the numerical values; row 293 (“Table II”) arabic numbers; the fluence value must be written directly on the SEM images (Figure 3).

2. Rows 85-87 (“Moreover, post-annealing treatment or high deposition temperature (200 °C) is required, in order to eliminate the oxygen contamination and improve the quality of the film surface [10].”) I cannot agree with the idea specified in the text, as the deposition of films in ns-PLD requires high temperatures and/or subsequent heat treatment. On the contrary, regardless of whether the laser is ns or fs, the typical PLD technique allows obtaining high-quality films at low substrate temperatures or even at room temperature. Instead, high temperatures and/or subsequent heat treatment are needed for the deposition of thin films by other techniques, such as chemical ones. In both regimes, ns and fs, through the laser ablation of a source material, a cloud of energetic particles is generated (with a plus for the one with a shorter laser pulse length) and high-quality films are obtained at low substrate temperatures. The cited reference [10] is not related to the idea specified in the text. Among the references that can be consulted are [Applied Surface Science 418 (2017) 456–462;].

3.      Rows 94-104, Paragraph presented here should be revised and corrected. It expresses experimental details that should be moved to corresponding section (the pulse length of 0.5 ps; investigated fluence range; the entire sentence “Depositions have been carried out on Si substrates by using boron targets with natural isotopic abundances (19.9 % of 10B and 80.1 % 11B).”).

4.      Rows 94 and 104; the presentation of the results in an explicit manner should not be done here, at the end of the introduction, and only those remarkable, that prove the sub-ps PLD technique is an efficient technique to obtain boron films with suitable properties for applications in the conversion of the thermal neutron’s technology.

5.      Rows 118-119 (“The silicon substrate was placed in the visible plume 118 cone formed during the ablation process, at a target-substrate distance d = 3 cm.”). If you had no other considerations that could be proven by measurements, then it should simply be mentioned that the depositions were made for a constant distance of 3 cm.

6.      The presentation of the experimental parameters in a table (see Table 1) does not bring more clarity and is redundant since most of the parameters have already been presented elsewhere. You should include all the information in the text. Revise and reformulate the captions of all presented tables. A large part of the results presented in Table 2 can be found later in Figures 1 and 2. They should be removed and presented only those not included in figures.

7.      In experimental section, the type of copper mesh and the provider must be specified. Also, it should be mentioned how the intimate contact with the substrate was ensured so that the deposition does not take place under the metal wires.

8.      In the paragraph located on rows 142-144 it is specified “The surface reflectance of boron target was experimentally determined by means of a UV-VIS Lambda 900 Perkin-Elmer spectrophotometer, equipped with a Labsphere integrating sphere.” but there are no results related to these measurements in manuscript.

9.      In manuscript, the term "fluence" is used, while in all figures "energy density" appears. For a better concordance between the discussion and the presented data, the same terms should be used. The legends must be indicated on all figures and also the fitting parameters in Figure 1.

10.  Rows 218-220, I believe you refer to Figure 2b (“From the fitted logarithmic curve (Fig. 1a, dashed line), the experimental value of the evaporated mass at the threshold laser fluence value (0.08 J/cm2) is about 8.6 219 ng/pulse.”).

11.  The sentence “The transition between the above two regimes as potential ablation mechanisms in metal by irradiation of sub-ps laser, is also confirmed by…” needs to be reviewed.

12.  The “forward-directed nature of the PLD technique” (rows 240-243) is not an argument to justify the non-uniformity in thickness of the deposited film. I would rather say that this non-uniformity in thickness and other properties of the deposited film that varies when going from the film center to its edge is due to the angular distribution of the plasma particles. The ion flux measurements for different angles with respect to the normal presented in  [Phys Rev Lett 84 (2000) 3998-4001] may be useful. Additional comments regarding the correlation of the angular distribution of the plasma particles and the nonuniformity of the film thickness should be made.

13.  In the paragraph marked at rows 260-264, it is presented the thickness variation with fluence. Then, in the next two rows a remark to the film thickness non-uniformity is made (“In these experiments, the non-uniformity of the film thickness is attributed mainly to the plume expansion geometry and in less extent to plume deflection effect [39, 40].”. Then the discussion turns back to thickness variation. That is confusing. It should be explained first why the thickness decreases with fluence (it may be a self-sputtering process as well). I believe that just this angular plume deflection with fluence is the reason for this unusual decrease in thickness. This non-uniformity in thickness that is a fingerprint of films deposited in PLD is due to the angular distribution of plasma particles. It has less influence on the thickness variation with fluence determined for a certain spatial point.

 14.  Rows 292-295, this paragraph must be reworded; it is not grammatically correct. Again, it is not necessary that the data presented in Fig. 3 (particle density distribution) also appear in Table 2. The results presented here should be included and discussed together with those presented in the previous paragraph. It must also be specified how the determination of this parameter was carried out.

15.  Rows 304-307, More recent articles can be consulted regarding the expansion velocities of plasma particles and the influence of energetic particles on the quality of films deposited in PLD [Spectrochimica Acta Part B: Atomic Spectroscopy 196 (2022) 106510] .

16.  Rows 309-312, the paragraph should be presented later and not before the presentation of the results related to the adhesion to the substrate and stability of the boron films.

17.  Rows, 321-324. Since the adhesion testing method does not allow obtaining a parameter that can be compared with the literature, it cannot be stated that the use of ps-PLD in the range of intensities mentioned improves the adhesion of the films to the substrate. It can be stated instead that the use of the ps-PLD technique in the range of laser radiation intensities leads to the production of boron films of good quality with good adhesion to the substrate.

18.  Rows 324-325. Again, no parameter was compared with films obtained by other techniques, so the obtained results demonstrate only that the PLD technique is one of the promising techniques for obtaining boron films that overcome adhesion problems.

19.  The sentence “All the sub-micrometer droplets, observed on the B film surface, have a circular geometrical shape with flattened morphology. All this suggests that the ablated material is ejected in liquid phase from the target surface” is not a concluding remark; please remove or reword it. Next sentence (rows 328-333) can be used in the statement of a conclusion regarding the surface quality of the obtained film. It is not appropriate in the actual form. Rows 338-342, the paragraph must be reworded; it is ambiguous. It should also be specified in the experimental parameters section what kind of target was used in this study, 10B-enriched target or one with 10B isotope in natural concentration.

Author Response

Dear Reviewer,

Please find the revisions to manuscript (materials-2164615 ) “Sub-ps pulsed laser deposition of boron films for neutron detector applications”, by Maria Luisa De Giorgi, et al. 

We thank you for the positive evaluation our research and for the valuable advices to improve our paper. In order to address the requested revisions we modified the paper in several points, which are evidenced in the manuscript in red colour.

Reviewer 2

The manuscript submitted by Giorgi et al. ("Sub-ps pulsed laser deposition of boron films for neutron detector applications") focuses on the deposition of boron films at room temperature on silicon substrate using the ps-PLD technique. The authors investigate the effect of laser fluence on the laser ablation process of the boron target and subsequently on the obtained film properties. The research is of interest since it addresses a current issue, namely the search for alternative methods that can be used in the manufacture of boron-based neutron detectors. However, the manuscript still needs a lot of work in terms of structuring, the presentation of additional experimental details, and lastly, the requirement for additional explanations of some of the obtained results. The remarks I would like to make are given below.

Comment 1: The manuscript contains some small inaccuracies that can be easily corrected such as: the "synthesis" term from the abstract should be replaced with deposition, fabrication, production, etc.; there is no consistency regarding the introduction and use of abbreviations (for example, the terms PLD, B film, fluence F from row 170 etc.); one single decimal should be used for the laser fluence estimates (“…laser fluence, ranging from 0.40 to 3.33 J/cm2”) and also specified in the experimental details section how the laser fluence was determined; the provider of the silicon substrate must be specified in experimental section; it is not necessary to mention all over the manuscript the fluence range, it can be simply stated "in the investigated fluency range" instead of the numerical values; row 293 (“Table II”) arabic numbers; the fluence value must be written directly on the SEM images (Figure 3).

Authors’ reply: According to the reviewer’s recommendation, in the abstract “synthesis” has been substituted with “deposition”, “F” has been substituted with “fluence” and “B” with “boron”.

To specify how the laser fluence has been determined in the section “Materials and Methods” we added the sentence: “calculated as the ratio between the laser energy impinging onto the target surface and the irradiated area” at rows 119-120.

Moreover, for the fluence values we have indicated only one single decimal according to the Reviewer’s suggestions.

Furthermore, when we refer to the whole fluence range instead to mention it by means of the numerical values, we used “in the investigated fluence range” as suggested by the Reviewer. Rows 286-287  

Finally, we used the Arabic numbers for the Tables and specified the provider of the silicon (“Goodfellow GmbH Supplier”, row 108-109)

Comment 2: Rows 85-87 (“Moreover, post-annealing treatment or high deposition temperature (200 °C) is required, in order to eliminate the oxygen contamination and improve the quality of the film surface [10].”) I cannot agree with the idea specified in the text, as the deposition of films in ns-PLD requires high temperatures and/or subsequent heat treatment. On the contrary, regardless of whether the laser is ns or fs, the typical PLD technique allows obtaining high-quality films at low substrate temperatures or even at room temperature. Instead, high temperatures and/or subsequent heat treatment are needed for the deposition of thin films by other techniques, such as chemical ones. In both regimes, ns and fs, through the laser ablation of a source material, a cloud of energetic particles is generated (with a plus for the one with a shorter laser pulse length) and high-quality films are obtained at low substrate temperatures. The cited reference [10] is not related to the idea specified in the text. Among the references that can be consulted are [Applied Surface Science 418 (2017) 456–462;].

Authors’ reply: We agree with the reviewer comments and to address his/her suggestion we have eliminated the sentence.

Comment 3: Rows 94-104, Paragraph presented here should be revised and corrected. It expresses experimental details that should be moved to corresponding section (the pulse length of 0.5 ps; investigated fluence range; the entire sentence “Depositions have been carried out on Si substrates by using boron targets with natural isotopic abundances (19.9 % of 10B and 80.1 % 11B).”).

Authors’ reply: We removed the laser parameters from the “Introduction” since they have been also indicated later in the “Materials and Methods” section and we moved the sentence “PLD experiments were carried out by using boron targets with natural isotopic abundances (19.9 % of 10B and 80.1 % 11B)….” from “Introduction” to “Materials and Methods”. Rows 103-108

Comment 4: Rows 94 and 104; the presentation of the results in an explicit manner should not be done here, at the end of the introduction, and only those remarkable, that prove the sub-ps PLD technique is an efficient technique to obtain boron films with suitable properties for applications in the conversion of the thermal neutron’s technology.

Authors’ reply: We removed the results from the “Introduction” and left only the most noteworthy comments.

Comment 5: Rows 118-119 (“The silicon substrate was placed in the visible plume 118 cone formed during the ablation process, at a target-substrate distance d = 3 cm.”). If you had no other considerations that could be proven by measurements, then it should simply be mentioned that the depositions were made for a constant distance of 3 cm.

Authors’ reply: In the text we left only “The silicon substrate was placed at a target-substrate distance d = 3 cm.Rows 115-116

Comment 6: The presentation of the experimental parameters in a table (see Table 1) does not bring more clarity and is redundant since most of the parameters have already been presented elsewhere. You should include all the information in the text. Revise and reformulate the captions of all presented tables. A large part of the results presented in Table 2 can be found later in Figures 1 and 2. They should be removed and presented only those not included in figures.

Authors’ reply: Table 1 has been removed and the information about “laser power density” and “background pressure” have been inserted in the “Materials and Methods” section. Furthermore, we revised all the captions of the other tables.

It’s true that the results reported in the table are used for the subsequent graphs, anyway for the sake of clarity we think they should be cited also in the table since it is not immediate to infer the values from the figures.

Comment 7: In experimental section, the type of copper mesh and the provider must be specified. Also, it should be mentioned how the intimate contact with the substrate was ensured so that the deposition does not take place under the metal wires.

Authors’ reply: It has been used a basic copper mesh one can find in a laboratory and it has been mechanically well fixed on the substrate by using mask and screws in order to ensure a close contact between substrate and mesh. To clarify this issue we added in the text the following sentence: “To avoid material deposition under the metal wires, the meshes have been mechanically well fixed on the substrates, by using mask and screws, ensuring a close contact between substrates and meshes”. Rows 125-128

Comment 8: In the paragraph located on rows 142-144 it is specified “The surface reflectance of boron target was experimentally determined by means of a UV-VIS Lambda 900 Perkin-Elmer spectrophotometer, equipped with a Labsphere integrating sphere.” but there are no results related to these measurements in manuscript.

Authors’ reply: The measured reflectance is reported in the last row of Table 2 and used to calculate laser fluence threshold, Fth, in formula (2).

Comment 9: In manuscript, the term "fluence" is used, while in all figures "energy density" appears. For a better concordance between the discussion and the presented data, the same terms should be used. The legends must be indicated on all figures and also the fitting parameters in Figure 1.

Authors’ reply: We thank the reviewer for the suggestions to improve the quality of the figures.We have rearranged all of them. 

Comment 10: Rows 218-220, I believe you refer to Figure 2b (“From the fitted logarithmic curve (Fig. 1a, dashed line), the experimental value of the evaporated mass at the threshold laser fluence value (0.08 J/cm2) is about 8.6 219 ng/pulse.”).

Authors’ reply: The fitted logarithmic curve correctly refers to Fig. 1a, dashed line.

Comment 11: The sentence “The transition between the above two regimes as potential ablation mechanisms in metal by irradiation of sub-ps laser, is also confirmed by…” needs to be reviewed.

Authors’ reply:

In order to clarify this aspect the sentence has been reviewed: “The transition between the above two regimes as potential ablation mechanisms in metal by irradiation of sub-ps laser is characterized by the presence of flakes and particulates on the film surface, as also indicated by recent research studies of Li and Guan [38]. The same evidence has been observed in our current work at 3.3 J/cm2.” Rows 236-240

Comment 12: The “forward-directed nature of the PLD technique” (rows 240-243) is not an argument to justify the non-uniformity in thickness of the deposited film. I would rather say that this non-uniformity in thickness and other properties of the deposited film that varies when going from the film center to its edge is due to the angular distribution of the plasma particles. The ion flux measurements for different angles with respect to the normal presented in  [Phys Rev Lett 84 (2000) 3998-4001] may be useful. Additional comments regarding the correlation of the angular distribution of the plasma particles and the nonuniformity of the film thickness should be made.

Authors’ reply: To avoid misunderstanding, the sentence has been partially revised: “The thickness (Th) of the deposited film is not perfectly uniform at the entire substrate and in particular it decreases slightly towards the region farthest from the film centerrows 248-249…Moreover the non-uniformity in thickness of the deposited film is explained at rows 255-257: “In these experiments, the non-uniformity of the film thickness is attributed mainly to the plume expansion geometry, to the angular distribution of plasma particles and in less extent to plume deflection effect [39, 40]. ”.

To support our statement, we replaced the old reference n. 39 with the reference suggested by the reviewer, which is more appropriate.  [B. Toftmann, J. Schou, T.N. Hansen, J.G. Lunney. Angular distribution of electron temperature and density in a laser-ablation plume. Phys. Rev. Lett. 84 (2000) 3998-4001]

Comment 13: In the paragraph marked at rows 260-264, it is presented the thickness variation with fluence. Then, in the next two rows a remark to the film thickness non-uniformity is made (“In these experiments, the non-uniformity of the film thickness is attributed mainly to the plume expansion geometry and in less extent to plume deflection effect [39, 40].”. Then the discussion turns back to thickness variation. That is confusing. It should be explained first why the thickness decreases with fluence (it may be a self-sputtering process as well). I believe that just this angular plume deflection with fluence is the reason for this unusual decrease in thickness. This non-uniformity in thickness that is a fingerprint of films deposited in PLD is due to the angular distribution of plasma particles. It has less influence on the thickness variation with fluence determined for a certain spatial point.

Authors’ reply: To address this issue we reorganized the paragraph moving the sentence “In these experiments, the non-uniformity of the film thickness is attributed mainly to the plume expansion geometry, to the angular distribution of plasma particles and in less extent to plume deflection effect [39, 40].”( Rows 255-257) before the discussion about the dependence of thickness on fluence.

Comment 14:  Rows 292-295, this paragraph must be reworded; it is not grammatically correct. Again, it is not necessary that the data presented in Fig. 3 (particle density distribution) also appear in Table 2. The results presented here should be included and discussed together with those presented in the previous paragraph. It must also be specified how the determination of this parameter was carried out.

Authors’ reply: The paragraph has been reworded as follows: “The surface droplet density, by counting the number of the droplets over an area of about 120 μm2, has been estimated. The results, reported in Table 1 are plotted in Fig. 3 as a function of the laser fluence, in the range from 0.4 to 1.1 J/cm2. The graph shows that the surface droplet density grows with the fluence.” Rows 298-301

As regarding the presentation of the results in the table, even if we agree with the reviewer that it could seem redundant, we think that it should be helpful and appreciated by the reader.

Comment 15: Rows 304-307, More recent articles can be consulted regarding the expansion velocities of plasma particles and the influence of energetic particles on the quality of films deposited in PLD [Spectrochimica Acta Part B: Atomic Spectroscopy 196 (2022) 106510] .

Authors’ reply: we thank the reviewer for having recommended this recent article, which has been included in the reference list in place of the 42.

Comment 16: Rows 309-312, the paragraph should be presented later and not before the presentation of the results related to the adhesion to the substrate and stability of the boron films.

Authors’ reply: The sentence has moved at the end of the “Film adherence to substrate and stability” section and slightly modified in the following way:

“As a result, the PLD technique in sub-ps regime in peculiar experimental conditions is particularly suitable to prepare boron films to be used in thermal neutron detectors. Indeed, it must be emphasized, once again, that this tribological property of the films is strongly demanding for any kind of application”.  Rows 328-331

Comment 17: Rows, 321-324. Since the adhesion testing method does not allow obtaining a parameter that can be compared with the literature, it cannot be stated that the use of ps-PLD in the range of intensities mentioned improves the adhesion of the films to the substrate. It can be stated instead that the use of the ps-PLD technique in the range of laser radiation intensities leads to the production of boron films of good quality with good adhesion to the substrate.

Comment 18: Rows 324-325. Again, no parameter was compared with films obtained by other techniques, so the obtained results demonstrate only that the PLD technique is one of the promising techniques for obtaining boron films that overcome adhesion problems.

Authors’ reply to comments 17 and 18: According to the reviewer’s proposal we revised the sentence as follows:  

It has been proved and confirmed that the high kinetic energy of the ablated material produced in our experimental conditions (1012-1013 W/cm2), leads to the production of boron films of good quality with high adhesion to the substrate. Hence, the growth of boron thin films by PLD seems to be a very promising approach to overcome the adhesion problems.” Rows 335-339

Comment 19:

  1. a) The sentence “All the sub-micrometer droplets, observed on the B film surface, have a circular geometrical shape with flattened morphology. All this suggests that the ablated material is ejected in liquid phase from the target surface” is not a concluding remark; please remove or reword it. Next sentence (rows 328-333) can be used in the statement of a conclusion regarding the surface quality of the obtained film. It is not appropriate in the actual form.

Authors’ reply: The sentence has been reworded in this new form: “In the fluence range 0.4-1.1 J/cm2 the boron films are covered of sub-micrometer droplets with a circular geometrical shape and flattened morphology due to the fact that the ablated material is ejected in liquid phase from the target surface. Differently, the sample deposited at 3.3 J/cm2 shows a surface covered by flakes and fragments likely due to the transition of the ablation process from spallation regime to phase explosion regime.” Rows 340-344

  1. b) Rows 338-342, the paragraph must be reworded; it is ambiguous. It should also be specified in the experimental parameters section what kind of target was used in this study, 10B-enriched target or one with 10B isotope in natural concentration.

Authors’ reply: To be less ambiguous the paragraph has been rewarded as follows:

In future, starting with the results obtained in this work, we are planning to deposit films using 10B enriched targets to be tested as thermal neutron detectors. Indeed, even if the relatively high isotopic abundance of 10B in target of boron with natural isotopic concentrations could warrant the use of the deposited films for neutron sensitive converters, 10B-enriched targets will certainly improve the detection efficiency”. Rows 351-355

Moreover, in our work we used boron targets with natural isotopic abundances, as described in “Materials and Methods” : “PLD experiments were carried out by using boron targets with natural isotopic abundances (19.9 % of 10B and 80.1 % 11B).” Rows 103-104

Best Regards

Maria Luisa De Giorgi

Reviewer 3 Report

In this work, authors report the synthesis of boron films on silicon substrates, by using sub-picosecond pulsed laser deposition (PLD), at room temperature. The role of the energy density on the film thickness and uniformity on substrate was investigated by SEM analyses, with laser fluences in the range from 0.40 to 3.33 J/cm2, being the purpose of this research to find out a reliable and reproducible method, leading to the fabrication of thick boron coatings suitable for neutron detection technology.

The work is of current interest and I found that the experimental procedure to stablish the parameters for laser usage are accurate and well presented.

I suggest that authors must make any comment about the deposited material, as not elemental/chemical results are included, not even EDS from SEM:

i.                    How is assured that films is almost boron and how to consider what traces of oxygen (or others adsorbed elements) are in final samples?

ii.                  The authors even mention that PLD can induce oxide formation, as a result of the interaction of the target with the plume and the laser pulses, do they discard that these interacting processes did not altered the chemical composition of boron films? Explain please

Author Response

Dear Reviewer,

Please find the revisions to manuscript (materials-2164615 ) “Sub-ps pulsed laser deposition of boron films for neutron detector applications”, by Maria Luisa De Giorgi, et al.  

We thank you for the positive appraisal of our work.and in order to address the requested revisions we modified the paper in several points, which are evidenced in the manuscript in red colour.

Reviewer 3

In this work, authors report the synthesis of boron films on silicon substrates, by using sub-picosecond pulsed laser deposition (PLD), at room temperature. The role of the energy density on the film thickness and uniformity on substrate was investigated by SEM analyses, with laser fluences in the range from 0.40 to 3.33 J/cm2, being the purpose of this research to find out a reliable and reproducible method, leading to the fabrication of thick boron coatings suitable for neutron detection technology.

The work is of current interest and I found that the experimental procedure to stablish the parameters for laser usage are accurate and well presented.

Comments

I suggest that authors must make any comment about the deposited material, as not elemental/chemical results are included, not even EDS from SEM:

  1. a) How is assured that films is almost boron and how to consider what traces of oxygen (or others adsorbed elements) are in final samples?
  2. b) The authors even mention that PLD can induce oxide formation, as a result of the interaction of the target with the plume and the laser pulses, do they discard that these interacting processes did not altered the chemical composition of boron films? Explain please

Authors’ reply: During the SEM observations, elemental analyses through EDX technique have been performed too, even if we did not report this information in the manuscript for the sake of brevity.

From the EDX spectra we inferred that the deposited films were composed essentially by boron atoms. However, in the spectra we also observed the oxygen and carbon peaks, indicating that the surface was polluted by oxygen and by carbon (with an atomic concentration below 10%), presumably coming from environmental and vacuum chamber residual gas.  

Now we have added this data also in the text:

During the SEM observations, elemental analyses through EDX technique have been performed, too. From the EDX spectra we inferred that the deposited films were composed essentially by boron atoms. However, in the spectra we also observed the oxygen and carbon peaks, indicating that the surface was polluted by oxygen and carbon (with an atomic concentration below 10%), presumably coming from environmental and vacuum chamber residual gas”. Rows 302-307

Moreover,in “Materials and Methods” we included the information about the instrumentation used for EDX analyses: “The chemical composition of the films was deduced by EDX analysis (OXFORD INCAPenta FETx3). The accuracy of the measurements was about 15%.” Rows 134-136

Finally, the section “3.2.2. Film surface morphology” has been renamed as “3.2.2. Film surface morphology and composition”.

Best Regards

Maria Luisa De Giorgi

Round 2

Reviewer 2 Report

The manuscript can be accepted in its current form.